# A Degraded Finger Vein Image Recovery and Enhancement Algorithm Based on Atmospheric Scattering Theory

**DOI:** 10.3390/s24092684

**Published:** 2024-04-24

**Authors:** Dingzhong Feng, Peng Feng, Yongbo Mao, Yang Zhou, Yuqing Zeng, Ye Zhang

**Affiliations:** 1College of Mechanical Engineering, Zhejiang University of Technology, Hangzhou 310014, China; fdz@zjut.edu.cn (D.F.); 2112102125@zjut.edu.cn (P.F.); 1121702012@zjut.edu.cn (Y.M.); 2112102576@zjut.edu.cn (Y.Z.); 2112102570@zjut.edu.cn (Y.Z.); 2Zhejiang Jinghong Intelligent Technology Co., Ltd., Lishui 321400, China

**Keywords:** finger vein image recognition, atmospheric scattering theory, gray value reconstruction, image recovery, image enhancement

## Abstract

With the development of biometric identification technology, finger vein identification has received more and more widespread attention for its security, efficiency, and stability. However, because of the performance of the current standard finger vein image acquisition device and the complex internal organization of the finger, the acquired images are often heavily degraded and have lost their texture characteristics. This makes the topology of the finger veins inconspicuous or even difficult to distinguish, greatly affecting the identification accuracy. Therefore, this paper proposes a finger vein image recovery and enhancement algorithm using atmospheric scattering theory. Firstly, to normalize the local over-bright and over-dark regions of finger vein images within a certain threshold, the Gamma transform method is improved in this paper to correct and measure the gray value of a given image. Then, we reconstruct the image based on atmospheric scattering theory and design a pixel mutation filter to segment the venous and non-venous contact zones. Finally, the degraded finger vein images are recovered and enhanced by global image gray value normalization. Experiments on SDUMLA-HMT and ZJ-UVM datasets show that our proposed method effectively achieves the recovery and enhancement of degraded finger vein images. The image restoration and enhancement algorithm proposed in this paper performs well in finger vein recognition using traditional methods, machine learning, and deep learning. The recognition accuracy of the processed image is improved by more than 10% compared to the original image.

## 1. Introduction

With the development of big data and information technology, there is an increasing demand for a higher level of identification technology. Finger vein recognition is a live detection technology with the following advantages. (i) Finger vein lines are unique and unchanging in every individual [1]. It has been medically proven that the vein pattern in the fingers of each individual is specific and unique. Even in twins, there are some differences in the distribution of the veins. And the shape stabilizes with age. (ii) Finger vein images can only be acquired from a healthy human body with good blood circulation. This makes it extremely difficult to steal and copy finger vein images. (iii) Finger vein recognition is resistant to interference. Unlike fingerprints, palm prints, etc., finger vein recognition technology collects information about the distribution of veins inside the finger. This information is not affected by factors such as finger surface stains and breakage [2]. Based on the above features, finger vein recognition technology is secure and highly anti-counterfeiting.

The principle of finger vein recognition technology lies in the fact that, when near-infrared (NIR) light is used to penetrate the finger, the area in which it is located has a smaller gray value because the hemoglobin in the vein region within the finger has a stronger absorption capacity for NIR light. This region is also known as the venous region. The non-vein region of the finger absorbs less infrared light, resulting in a higher gray value in this region. This region is known as the non-vein region. By comparing the different gray values of the vein region and non-vein region in the finger vein image, the unique vein pattern and topology of the finger can be reflected. Thus, the finger vein images described above can be used for identification [3]. The system used by the project team to collect the finger vein image dataset (ZJ-UVM) is shown in Figure 1. The schematic and device diagrams of the system are given in Figure 1a,b. In order to ensure the operation visualization and recognition effect during the acquisition process, the device is set up as a semi-open type. The light source is placed on both sides and tilted at 60 degrees to penetrate the finger. Figure 1c gives the finger vein image after the region of interest (ROI) is processed. It can be seen in this figure that the vein vessels will form a stripe projection on the image. The region of the finger vein stripe projection has a lower gray value, which is the vein region mentioned above.

The images captured by existing finger vein devices often suffer from severe image quality degradation. This affects the accuracy of the recognition to some extent. For some individuals, finger vein recognition is not even possible because the vein pattern of their finger vein image is not distinct enough. Even re-capture cannot solve this problem. The reason for this is that there is inherent scattering, uneven light intensity, and near-infrared light attenuation during finger tissue imaging when acquiring vein images [4]. This results in unclear vein patterns in the images and a low contrast between vein and non-vein regions. The specific manifestations on these images are: first, the vein edges are blurred, and there is no obvious difference between the vein regions and the bordering non-vein regions. Second, in some images, the non-vein regions of some fingers will have smaller gray values than some vein regions. The extracted vein texture features are too thick or too thin and deviate from the actual veins. Third, the vein regions are difficult to determine and extract directly. Therefore, restoring and enhancing original finger vein image texture features is a key step in finger vein authentication and recognition techniques.

To solve the above problems, this paper examines and studies the process by which the internal tissues and skeleton of the finger absorb near-infrared light. It is found that the local imaging process of the image is basically the same as the process of capturing an image in bad weather. Therefore, we propose a finger vein image recovery and enhancement algorithm based on atmospheric scattering theory in computer vision. The enhancement of the studied finger vein images is recognized through experiments.

The main contributions of this paper are as follows:(i)We improve the gamma transform method for use in the conversion of gray values of finger vein images. According to the image gray level requirement, the image gray value is normalized to within a specific threshold value. It is used to improve the gray level difference of pixel points at the vein edges. This method avoids the problem that gray values cannot be reconstructed in the next step because they exceed a fixed threshold;(ii)The main points of the image gray value reconstruction method designed in this paper include (1) proposing image local-region light intensity for eliminating the differences in incident light intensity between different regions due to different tissues and skeletons. (2) Designing a pixel mutation filter to solve the problem of different absorption rates of light intensity in local regions, and realizing the segmentation of vein and non-vein regions. We combine the above method points with the atmospheric scattering theory formula to realize finger vein image gray value reconstruction;(iii)The homomorphic filtering technique is introduced to globally normalize the image by decreasing the gray value of the high-frequency components in the vein regions and increasing the gray value of the low-frequency components in the non-vein regions. This can further improve the contrast of the image.

The rest of the paper is organized as follows: Section 2 presents work related to this paper. Section 3 describes the atmospheric scattering model. Its application to finger vein images is also analyzed and studied. Section 4 presents the framework and details of the image restoration and enhancement algorithm of this paper. In Section 5, we analyze and experimentally verify the effectiveness of the application of the algorithm in this paper. Finally, the conclusion of this paper is given in Section 6.

## 2. Related Works

In order to solve the problem of the degradation of finger vein images, which leads to decreasin of recognition accuracy, the related finger vein image recovery and enhancement algorithms are mainly researched based on the characteristics of the finger vein image itself and the depth features.

### 2.1. Finger Vein Image Enhancement Based on Optical Features

From the optical features of finger vein images, Lee et al. [5,6] performed deconvolution based on the amount of finger vein image blur. And the image gray level was corrected using least squares filtering, which was used to recover the clear finger vein image. Yang et al. [7] proposed a bio-optical model (BOM) specific to finger vein imaging. The light-scattering component was estimated and correctly removed by the ensemble. Their method realizes the recovery of finger vein images with a simple computational procedure. However, the method is not effective for severely degraded finger vein images. Guo et al. [8] and others used methods such as grayscale gradient and region enhancement for image recovery based on this. Zhang et al. [9] proposed an image enhancement method based on weighted wavelet visual perception fusion. By utilizing the weighted wavelet visual perception fusion strategy, enhanced global and local images were fused to obtain high-quality images. However, the fuzzy kernel parameters of the above methods are difficult to determine due to the differences between different finger vein images. More importantly, the recovery effect of the above methods is overly dependent on the estimation and setting of parameter values. When replacing different datasets, the difference in the processing effect is large. Therefore, these methods need to be improved.

In order to solve the scattering problem of light in the propagation process, Kaiming et al. [10] proposed a classical dark-channel a priori algorithm. In their algorithm, the thickness of the haze in the image is estimated, followed by image enhancement based on atmospheric scattering theory. Lee et al. [11] proposed the use of dark-channel a priori knowledge of hyperpixels to determine the absorption and scattering rates of light, and combined it with continuous color correction to achieve image enhancement. Zhang et al. [12] first corrected different color channel color biases. Then, different enhancement methods were used according to the characteristics of different channel layers. Thus, the image contrast was improved and enhanced in detail. Further, while the images captured by the finger vein device in their study are formed by the transmission or reflection of near-infrared light, and this type of image has the same gray value for each channel. This makes it so that there is no need for dark-channels or special features of different channels. Zhang [13] proposed a local adaptive contrast enhancement technique for the problem of chromatic aberration and low visibility of images. It effectively improves the performance of underwater image algorithms. Bhateja [14] proposed a Multi-scale Retinex with Chromacity Preservation (MSRCP) to improve the contrast of images. Then, Li et al. [15] developed an adaptive multi-scale retinal (AWRM) method. In this method, the chunked images are processed separately and combined globally using optimal scale parameters and weights. Through experiments, a better experimental performance was obtained. The above methods have achieved some results on three-channel images. However, the processing effect on single-channel images such as finger vein images still has some shortcomings.

### 2.2. Finger Vein Image Enhancement Based on Texture Features

Liu et al. [16] proposed fully extracting finger vein features using global histogram equalization. Chen et al. [17] proposed a histogram equalization algorithm that fuses global and local finger vein images based on the distribution specificities of different regions the images. Lu et al. [18] proposed a modular adaptive histogram equalization algorithm. Thair et al. [19] proposed a method combining image descriptors and global histogram equalization to improve the contrast of finger vein images. The method achieves non-returnable image information and provides good security. The above method transforms the histogram distribution of gray values of an image as a whole and locally. The gray level difference between the vein region and the non-vein region of the image is enlarged to highlight the vein region and enhance the contrast of the image. However, these methods enhance the vein region along with the noise information. Due to the fragmented distribution of noise, it creates a large obstacle for the subsequent feature extraction process.

The Gabor filter can obtain the maximum possible joint resolution in both spatial and spatial-frequency domains and is gradually becoming widely used for finger vein image enhancement. Kumar et al. [20] used the Gabor filter and morphological transformation to obtain binary images for image matching. However, in the enhancement process, the fake veins generated by irregular shadows are also enhanced and, hence, the accuracy of the recognition is not high. Zhang et al. [21] proposed an adaptive Gabor filter incorporating a convolutional network to enhance the vein region and suppress pseudo-veins to overcome this problem. However, this method did not completely solve the above problem. Subsequently, Menaha et al. [22] applied direction-specific Gabor filters in their work, which were convolved with a scaled-up picture to filter out unnecessary areas. Kovac [23] used a 2D Gabor filter to obtain the orientation and phase features of veins from finger vein images for recognition, and its parameters were locally adjusted to fit the local orientation and frequency of the underlying vein patterns. Zhang et al. [24] proposed a finger vein image enhancement algorithm based on a bootstrapped tri-Gaussian filter. Better results were achieved in maintaining the vein structure, local processing, and noise suppression. These methods have achieved good results in highlighting vein structures. However, due to the uneven grayscale distribution of some finger vein images, and even the presence of irregular shadow regions [25], the possibility exists of over-segmentation of the vein region when using such algorithms for enhancement [3].

### 2.3. Finger Vein Image Enhancement Based on Deep Feature

With the development of computer vision, neural networks are more and more widely used in the field of image processing. Pathak et al. [26] proposed a CNN image restoration model based on contextual coding in which semantic restoration was achieved by reconstructing the standard pixel loss and adversarial loss to construct image gray values. Du et al. [27] and Shand et al. [28] used feature semantic compression and fusion to construct a feature network for image recognition, and achieved quite good restoration results on large datasets. However, the finger vein image database that such methods are applied to is small in size and low in resolution, and cannot meet the requirements of deep model training. Choi [29] constructed a lightweight adversarial neural network (GAN) for enhancing finger vein images to achieve image deblurring. He et al. [30] proposed an improved GAN method in which, by adjusting the loss weights, the contrast between vein and non-vein regions is improved. It is effective in recovering the optical blur in original finger vein images. Gao et al. [31] proposed an image-processing method for adaptive blurring. This method utilizes the adversarial game features of the generator and discriminator to obtain better deblurring results. However, important vein texture information is largely lost due to the frequent downsampling process during the feature recovery and enhancement. This leads to poor enhancement of the image, especially for low-resolution and severely degraded images. Moreover, in practical applications, neural networks require advanced equipment platforms due to their complex structure and parameter requirements.

In order to obtain clear underwater images, Li et al. [32] combined the physical model of imaging with underwater optical features to realize image enhancement, and also designed a lightweight convolutional neural network model (CNN) for adapting different scenes. However, finger vein images are generally acquired using transmitted light, and the acquired light is affected by both scattering and the tissue structure and cannot be directly migrated for use. Hsia et al. [33] proposed a deep learning (DL) based semantic segmentation method and used the maximum curvature method to extract vein features. A better equal error rate was obtained. Li et al. [34] designed a finger vein model (FV-ViT) based on a visual transformer (ViT). Excellent recognition performance was achieved after adding strict regularization. Devkota et al. [35] integrated the DenseNet model, squeeze excitation (SE), and hybrid pooling (HP). Then, a series of preprocessing methods were used to obtain vein patterns. The method acquired better accuracy and generalization in the dataset. However, the above methods require advanced equipment and do not take into account the characteristics of the finger vein images themselves. This is not effective for the processing of heavily degraded images.

## 3. Image Recovery Model

In foggy and hazy environments, images acquired by cameras often suffer from severe scattering and degradation, resulting in extremely blurred contour edges between different objects. Narasimhan [36] proposed an atmospheric scattering model for processing single-channel transmission images to solve the degradation problem of monochrome images. After testing, the method provides good recovery of images in hazy environments. The total irradiance *E* received by the sensor is equal to the sum of the attenuation model *E_dt_* and the atmospheric light model *E_α_*:*E* (*d*, *λ*) = *E_dt_* (*d*, *λ*) + *E_α_* (*d*, *λ*)(1)

The attenuation model *E_dt_* refers to the way in which the light reflected from the scene is attenuated as it propagates to the camera. The attenuation irradiance is described as:*E_dt_* (*d*, *λ*) = *E*_∞_(*λ*) *r*(*λ*) *e*^−*θ*(*λ*)^*^d^ d*^−2^(2)
where *d* refers to the width from the field point to the camera and *λ* represents the wavelength. *θ*(*λ*) is called the atmospheric scattering coefficient, and its magnitude reflects the ability of the atmosphere to scatter light per unit volume. *E*_∞_ refers to the brightness of the light, and in atmospheric modeling, it refers to the horizon brightness. *r* is a function of the reflectivity properties and sky aperture, describing the scene point.

The atmospheric light model *E_α_* refers to the total irradiance of ambient light that enters the camera through atmospheric multiple scattering, refraction, etc., coinciding with a scene point. This term reflects the light intensity of the atmospheric light. *E_α_* can be described as:*E_α_* (*d*, *λ*) = *E*_∞_(*λ*)(1 − *e*^−*θ*(*λ*)^*^d^d*^−2^)(3)

In the acquisition of finger vein images, near-infrared light is transmitted through the finger. Based on the fact that hemoglobin is more sensitive to infrared rays, an area with a lower gray value on the captured image is determined to be a vein. This is followed by identification of the individual based on the pattern composed of the vein area as a characteristic of the individual. However, scattering and light intensity attenuation inevitably occur during the penetration of near-infrared light into the finger. Shown in Figure 2a is a schematic diagram of the main internal structures that affect the quality of finger vein imaging, including skin tissue, flesh, bone, and finger veins. The finger veins are located throughout the skin tissue and muscles. There are some differences between tissues and bones at different locations, and the further the distance between two pixels points, the greater the difference. This is a key factor that directly affects finger vein imaging. Figure 2b represents the schematic diagram of two-finger-vein-point imaging. The different lines in the figure indicate the different propagation modes of the light: NIR refers to the light that comes directly from the light source to the finger; attenuated light refers to the light that is weakened after refraction and reaches the vein area; refractive light refers to the light that is refracted and does not reach the designated vein area. Refractive light is the biggest cause of decreased contrast between vein and non-vein areas. The extraneous light is light from non-vein areas that is refracted and reaches the vein area. The venous points in the figure are located at different locations and the degree of attenuation varies greatly. This is the reason why the gray value of some vein areas is larger than that of non-vein areas (under normal circumstances the vein areas are smaller than the non-vein areas).

Unlike the scattering of particles in the atmosphere, there is complex multiple scattering in the finger vein imaging modes [7]. And since it cannot be directly ignored as in Equation (1), it is not possible to apply the theoretical model of atmospheric scattering directly to finger vein image recovery. The difference between the imaging of finger veins and atmospheric light imaging modes is:(i)Scene imaging in atmospheric systems is an incident imaging mode. The pixel values of the image are derived from light reflected from the object and atmospheric light from the environment. The atmospheric light in the environment remains essentially the same without interference, which is the key to the de-fogging algorithm. Therefore, the foggy image is affected by the severe scattering caused by the fog. However, a particular scenario is affected to the same degree. Finger vein imaging is a type of transmission imaging. The pixel values of the image are derived from the attenuation of a beam of light after it has passed through the finger. And the gray value of each pixel point is affected by the surrounding area. The degree of influence of the light is related to the light intensity and distance of the surrounding area. This means that different pixel locations are affected differently.(ii)Imaging environments have different scene depths. Imaging within atmospheric systems has a wide field of view. Therefore, the atmospheric scattering coefficient of a scene can be calculated based on the wavelength and scene depth. Finger vein imaging environments have a camera distance of less than 10 cm from the finger NIR light source. The effect of different regions cannot be described simply by a fixed function.(iii)In an atmospheric imaging system, the intensity of the light arriving at a scene is the same. It is absorbed by the scene points and reaches the camera. The finger vein device uses a string of LED light beads to provide near-infrared light. The intensity of the light entering the finger is different due to the different positions and thickness of the finger. And since the finger veins are not distributed in a single plane, the light intensity reaching different finger veins also varies.

Considering the different effects of the light attenuation rate, incident light intensity, and surrounding area, before introducing the image recovery and enhancement algorithms in this paper, we need to propose the following two processing methods based on the distribution characteristics of finger vein images to facilitate the processing:(1)Due to the more complex organization within the finger, different regions have different attenuation rates. However, as the region gets progressively smaller at each successive pixel point, the attenuation rate within each region is approximately the same. This value can be considered as a constant.(2)In the finger vein image, the gray value of the vein region in any small area is smaller than the gray value of the non-vein region. And there is a continuous gradient change in the gray value connecting the vein region and the non-vein region. These features are maintained even in heavily degraded finger vein images. Therefore, in a region larger than the width of the finger vein, when the gradient of the gray value changes continuously and exceeds a certain threshold, it can be recognized that there are both vein regions and non-vein regions. Conversely, it is possible in this scenario that there are only non-vein regions.

Currently, in the field of computer vision, in order to simplify the parameterization of a scattering model, the model for a single image can be described as:(4)$$I\left(x\right)=e^{-\mu D\left(x\right)}J\left(x\right)+I_{s}\left(1-e^{-\lambda D\left(x\right)}\right)$$
where *I* is the acquired image irradiance. *μ* denotes the extinction coefficient (set to constant in the calculation). Its magnitude determines the degree of image degradation. *D*(*x*) denotes the distance coefficient. *J* is the intrinsic intensity of the scene itself. *I_s_* is the irradiance of the light source, which is generally expressed as the highest brightness point of the scene [37]. Given *µ*, *D*(*x*), and *Is*, we can obtain the image’s irradiance without scattering. However, due to the narrow and complex environment of finger vein imaging, it is not possible to obtain these parameter values simply and accurately.

Based on the above analysis, the difference between the atmospheric imaging system and finger vein imaging lies in the atmospheric irradiance *Is* and the amount of light scattering. Therefore, in this paper, the image scattering model in finger veins is described as:*I*(*x*) = *J*(*x*) *ε*(*x*) + *A*(1 − *ε*(*x*))(5)
where *J*(*x*) is the light intensity of the image to be recovered, indicating the recovered image. *I*(*x*) is the acquired finger vein image. *ε*(*x*) is the absorptance of a pixel point of the finger, which is used to calculate the scattered light intensity and received light intensity of the light intensity of each pixel point. *A*(*x*) is the intensity of the scattered light in the localized region where the pixel point is located. In this paper, *A*(*x*) is a variable whose size is related to the light intensity in the region of *I*(*x*). At the time of acquisition, the natural light source is negligible, so the intensity of this region can also be considered as the sum of the scattered intensities of different particles in the region.

## 4. Image Recovery Algorithm

In Section 3 of this paper, it is shown that the acquired finger vein images are obtained by a combination of a light source, tissue transmittance, and scattering. This leads to the severe degradation of images. And the complex internal structure of the finger makes it difficult for many image enhancement algorithms to achieve the desired results. Therefore, we propose a combined local and global finger vein image recovery and enhancement algorithm based on atmospheric scattering theory.

The overall flowchart of the algorithm of this paper is shown in Figure 3. The image input is first processed with Gaussian blurring. And the finger vein image grayscale threshold is adjusted to the specified range by using the improved Gamma transform method. Then, the image gray value is reconstructed based on atmospheric scattering theory. Therefore, the local vein region and non-vein region are separated. Finally, the global normalization of the image is realized based on homomorphic filtering. The specific process of the algorithm is described in detail below:

### 4.1. Grayscale Value Quantization

Finger vein images can be influenced by the NIR luminosity. Different locations of the finger have different absorption rates. This leads to excessive differences in the overall brightness of the finger vein image. That is, the gray value of some areas is too large or too small. This greatly affects the subsequent reconstruction of gray values. Therefore, it is necessary to normalize the gray values of vein regions and non-vein regions in different finger vein images so that they are within a certain threshold range. Wang [38] adjusted the image parameters using local Gamma transformation and verified the effectiveness of the method. Its standard formula is:*I*(*x*, *y*) = (*Y*(*x*, *y*))*^γ^*(6)
where *I*(*x*, *y*) is the corrected gray value. *Y*(*x*, *y*) is the original image. *γ* is a control parameter, and when *γ* is less than 1 but greater than 0, the overall brightness increases. When *γ* is smaller than 1, the overall brightness decreases.

This method has satisfactory results for uniformly overexposed images. However, for a finger vein image, the overall gray value is generally too large or too small. The direct application of Gamma transformation cannot achieve effective separation. And it cannot be transformed to within a certain threshold. Therefore, we improved the gamma transform method to correct the image gray value. Firstly, the gray values of the finger vein images are normalized to the range [0, 1]. In order to improve the threshold range of different gray values after normalization and to differentiate between different sizes of gray values, this section describes the normalization of each image by its own maximum and minimum gray value as a base point. The transformation formula is:*G*0(*x*, *y*) = (*I*0(*x*, *y*) − *min*(*I*0))*/*(*max*(*I*0) − *min*(*I*0))(7)
where *G*0(*x*, *y*) is the corrected gray value. *I*0(*x*, *y*) is the original image’s gray value. *min*(*I*0) is the image’s minimum gray value. *max*(*I*0) is the image’s maximum gray value.

Next, Gaussian filtering is used to blur the normalized finger vein image to remove the noise and make the image smooth, with the processing equation:*Guess*(*x*, *y*) = (2*πσ*^2^)^−1^ *e*^0.5(*u*2+*v*2)2^*σ*^−2^(8)
*G*1(*x*, *y*) = *∑*(*Guess* (*x*, *y*) · *G*0(*x*, *y*))(9)
where *Guess*(*x*, *y*) is the Gaussian kernel and *σ* is taken to be 1.5. *G*1(*x*, *y*) is the pixel value after applying Gaussian blurring.

Finally, an exponential transformation is performed on the gray value of the image after the above processing. In order to ensure the threshold of the gray value of the image, upper and lower limits are added during the transformation process, and the final range of the gray value is [*G_m_*, *G_m_* + *G_n_*]. The calculation formula is:*Wg*(*x*, *y*) = *G_1_* (*x*, *y*)*^r^ G_m_* + *G_n_*(10)
where *Wg* is the transformed gray value and *r* is the index. When *r* > 1, the image’s gray value becomes smaller and the brightness becomes darker and vice versa. *G_m_* is the upper threshold limit and *G_n_* is the lower threshold limit.

### 4.2. Image Enhancement Based on Atmospheric Scattering Theory

In Section 3, by analyzing the atmospheric scattering theory, this paper gives the atmospheric scattering Formula (5) applicable to finger vein image enhancement. However, due to the complex internal structure of the finger, calculation of the absorption rate *ε* and light intensity *A* is very complicated. In this section, we process the above two parameters and get the final formula.

#### 4.2.1. Calculation of the Absorption Rate *ε*

To calculate the absorption rate *ε*, we assume that the optical intensity *A* is known. Therefore, *ε* is calculated as:*ε*(*x*, *y*) = *I_out_*(*x*, *y*)/*I_in_*(*x*, *y*)(11)
where *I_in_* is the incident light intensity and *I_out_* is the light intensity reaching the camera. In the imaging environment of finger veins, the sources of near-infrared light mainly include light sources and near-infrared wavelength light from natural light. However, compared to the NIR light source, the IR light in natural light is negligible. Therefore, the incident light intensity Iin is approximated as the light intensity *A* of the light source, i.e., *I_in_* = *A*.

The outgoing light intensity *I_out_* represents the part of the near-infrared light that is not absorbed, scattered, or reflected by the tissues of the finger and is captured by the camera, i.e., *I*_out_ = *I* (*I* is the image gray value). According to Equation (5), the outgoing light intensity is equal to the distinction between the gray value of the captured picture and the scattered component, i.e., *I* − *A*(*x*)(1 − *ε*(*x*)). However, the estimation of the scattering component of the grayscale image is difficult to obtain using practical calculations. From the perspective of image enhancement, atmospheric scattering theory is utilized to recover and enhance the image to improve the contrast between veins and non-veins. And the vein region and non-vein region are normalized to a certain threshold range to realize the segmentation of vein lines. This makes the vein lines clearer and the vein features easier to extract. Therefore, a filter of mutations pixel (FMP) is designed in this paper. The FMP can be described as:*FMP*(*x*, *y*) = *max_Ω_*_2_(*I*(*x*, *y*)) + *N·min*(*C_Ω_*_1_(*I*(*x*, *y*))*·*(*mx_Ω_*_1_ − *mx_Ω_*_2_), *mx_Ω_*_1_)(12)
*mx_Ω_* = *max_Ω_* (*I* (*x*, *y*))(13)
(14)$$N=\left\{\begin{array}{c} 1,\;\;\;C_{\Omega 1}(I(x,\;y))\;\geq \;\mathrm{count}{(}_{\Omega 2})/\mathrm{count}{(}_{\Omega 1})\\ -1,\;C_{\Omega 1}(I(x,\;y))\;<\;\mathrm{count}{(}_{\Omega 2})/\mathrm{count}{(}_{\Omega 1}) \end{array}\right.$$
where *FMP*(*x*, *y*) is the filtered pixel value. *Ω* is the gray value region of the image. (*x*, *y*) denotes the position of the pixel point and (*x*, *y*) ∈ *Ω*_2_ and *Ω*_2_ ∈ *Ω*_1_. *max_Ω_*(*I*(*x*, *y*)) is the maximum value in the region of *Ω*, and *min*() takes the minimum value. *count*(*Ω*) is the number of pixel points in the region of *Ω*. *C_Ω_*_1_(*I*(*x*, *y*)) is the ratio of the number of pixel points that are not larger than the *I*(*x*, *y*) to the number of all pixel points in the region. Therefore, the absorption rate *ε* can be described as*ε*(*x*, *y*) = *FMP*(*x*, *y*)/*I_in_*(*x*, *y*)(15)

#### 4.2.2. Calculation of the Area Light Intensity *A*

In a finger vein imaging environment, the near-infrared light source is the main factor affecting the image. This is because the near-infrared wavelength brightness of natural light is negligible compared to near-infrared light sources. Infrared light sources commonly used in finger vein collection devices include both top-emitting and side-emitting types. When infrared light is irradiated to the finger, on the one hand, there is some difference in the initial light intensity of the finger due to the uneven distribution of the light source and the shape of the finger. On the other hand, the organizational structure of different regions of the finger varies greatly, and even in the same non-vein region, the absorption rate of infrared light is not the same in different locations. This results in the gray value of some non-vein regions being smaller than that of the vein regions.

As shown in Figure 4, we selected two images of finger veins with severe degradation and counted the average gray values of the three regions, respectively. Among them, ① = 101, ② = 112, ③ = 109, ④ = 34, ⑤ = 65, and ⑥ = 14. By comparison, we found that the average gray value of different regions is directly related to the light source and the internal organization of the finger. The presence of veins has little effect on the average gray value. Therefore, it is difficult to extract the finger vein features of such fingers. Moreover, there is a large difference in the intensity of incident light reaching the finger veins, which makes it difficult to segment the vein region from the non-vein region.

Therefore, the concept of regional light intensity in finger vein images is proposed in this paper. In order to ensure the consistency of the gray value of recovered finger vein images, when applying Equation (5) to calculate the gray value of a pixel point, the average of the top 5% of points with larger gray values in the region is taken as the *A* value of the region, which is denoted by *A*(*x*, *y*). The formula is:*A*(*x*, *y*) = *maxi_Ω_*_1_(*I*(*x*, *y*))(16)
where *A* (*x*, *y*) is the regional light intensity at pixel point (*x*, *y*). *maxi_Ω_*_1_() is the average of the first 5% of *Ω*1 for regions with a large gray value.

Next, the finger vein image recovery algorithm according to the atmospheric scattering theory proposed in this section is applied to process the images and compare the changes in gray values before and after processing. As shown in Figure 5a,d, severe scattering of image I and image II occurs. Among them, the incident light intensity of image I Figure 5a is higher, and the overall gray value of the image is higher, whereas the incident light intensity of image II Figure 5b is lower and, hence, the overall gray value of the image is lower. The finger vein features of the two images are very blurred, and their direct application to identification will result in serious misclassification. Figure 5b shows the variation in the gray value of image I in the vertical direction. The gray value of each pixel point is derived from the average of three pixel points on the horizontal square. These three pixel point locations are the three pixels points with a horizontal pixel scale of [80, 82] at the same vertical height as the chosen pixel point. By comparison, it is found that the pixel values of the specified region change more gently, and the difference between the gray values of vein and non-vein is less than 50. Figure 5e is a graph of the variation in gray value in the vertical direction of image I. The gray value of each pixel point is derived from the average of three pixel points on the horizontal square. The gray value of its pixel points is obtained by the same method. The pixel value changes in the specified area follow a certain monotonicity, and the difference in the gray value of different pixel points in a small range is very small.

Figure 5c is a graph of gray value changes in the specified area of image I after processing. Figure 5f is the gray value change graph of image II after processing. The gray value of each point is obtained by the same method as described above. As can be seen from the figure, the trend of gray value changes in image I and image II after processing becomes obvious. And the gray values of pixels in small areas change drastically. This indicates that the contrast of the vein region of the image is significantly improved after enhancement using the method proposed in this section.

In Figure 6, a visualization of the grayscale values of the degraded finger vein image is shown. Figure 6a,c are the original images. Figure 6b,d are the processed images. The *x*-axis and *y*-axis represent the positions of pixel points. The height is the gray value of the pixel point. The higher the gray value, the higher the pixel point and the lighter the color. From the figure, it is known that the texture of the finger vein image is clearer after processing. And the difference in the size of the gray values between different veins is more obvious. This means that the texture characteristics of finger vein images are more obvious after processing.

### 4.3. Global Image Normalize

The finger vein image is processed as described above, which enhances the localized vein pattern contrast. However, since most of the previous processing enhancement takes into account the gray value changes in the current zone, it is necessary to continue the normalization process for the whole image. In order to solve the problem of the global normalization of finger vein images, this paper proceeds according to the basic principles of single-parameter homomorphic filtering [39] and designs a global normalization method based on homomorphic filtering.

Homomorphic filtering combines the irradiated and reflected components of an image in a multiplicative manner according to the image irradiation-reflection model. The descriptive equations are as follows:*f*(*x*, *y*) = *i*(*x*, *y*) *· r*(*x*, *y*)(17)
where *f*(*x*, *y*) is the acquired image. *i*(*x*, *y*) is the irradiation component, which represents the light portion. *r*(*x*, *y*) is the reflection component. Then, a logarithmic transformation is used to convert the product into a sum, which facilitates noise separation. The formula is as follows:*ln f*(*x*, *y*) = *ln i*(*x*, *y*) *· ln r*(*x*, *y*)(18)

Fourier transformation of Equation (18):*F*(*u*, *v*) = *I*(*u*, *v*) + *R*(*u*, *v*)(19)

The transformed results are processed using the transfer function *H*(*u*, *v*):*H*(*u*, *v*) *· F*(*x*, *v*) = *H*(*u*, *v*) *· I*(*u*, *v*) + *H*(*u*, *v*) *· R*(*u*, *v*)(20)
(21)$$H\left(u,\;v\right)=r_{H}-r_{L}\left[1\;-e^{-c{\left(\;\frac{D\left(u,\;v\right)}{{2D}_{0}}\right)}^{2n}}\right]+r_{L}\;$$
where *D*_0_ is the cutoff frequency and is taken as 25. *r_H_* is the high-frequency gain and is taken as 2. *r_L_* is the low-frequency gain and is taken as 0.5. *c* is the sharpening coefficient. n is the number of filter orders. *D*(*u*, *v*) is the distance from the frequency (*u*, *v*) to the center frequency (*u*_0_, *v*_0_) and is computed as:*D*(*u*, *v*) = ((*u* − *u*_0_)^2^ + (*v* − *v*_0_)^2^)^0.5^(22)

The unprocessed finger vein image is shown in Figure 7a. Figure 7b is the image in Section 4.2 after processing, which corresponds to the original image, and it effectively distinguishes the vein regions and non-vein regions in the image. However, since only the local gray value changes are considered in the processing, the difference between the vein and non-vein gray value thresholds in different regions is relatively large. Figure 7c shows the finger vein image after homomorphic filtering. The image vein texture of this image is clearer. The image contrast is further improved.

## 5. Experimental Results and Evaluation

In this section, we choose the publicly available dataset SDUMLA-HMT [40] and the ZJ-UVM finger vein dataset experimentally acquired by our group for our experiments. Among them, the SDUMLA-HMT dataset has 106 experimenters. Each experimenter provided six fingers. Six finger vein images were acquired for each finger. The total number of finger vein images in this dataset is 3816. The ZJ-UVM dataset is derived from finger vein images acquired using a device of our own design. The dataset consists of 73 experimenters. Each experimenter performed the same classification and image processing as described above. The dataset contains a total of 2628 finger vein images. In Figure 8, some of the finger vein images in the ZJ-UVM dataset are shown.

The algorithms for the experiments in this paper were written in Python 3.9. The computer equipment used was Intel^®^ Core^TM^ i7-9700F CPU @3.00 GHz with 32 GB of RAM. The operating system is Windows 10 64-bit Professional Edition.

### 5.1. Ablation Experiment

The degraded finger vein image recovery and enhancement algorithm based on atmospheric scattering theory proposed in this paper consists of a total of three steps: (1) Gray value metric; (2) image enhancement based on atmospheric scattering theory; (3) global image normalization. In order to verify the necessity of each step, ablation experiments are used.

A schematic of the ablation experiment is shown in Figure 9, where Figure 9a is the original finger vein image. These six images are the most severely degraded images in the SDUMLA-HMT dataset and the ZJ-UVM dataset. Figure 9b is the image processed by using the gray value metric and the enhancement algorithm based on atmospheric scattering theory. Through experiments, it is determined that the direct use of the enhancement algorithm based on atmospheric scattering theory leads to the gray value of the pixel points exceeding the threshold value (the gray value threshold value is 0~255). Figure 9c shows the finger vein image after processing using homomorphic filtering. Figure 9d is the image after processing using the degraded image recovery and enhancement algorithm based on atmospheric scattering theory proposed in this paper. From the figure, it is clear that processing with one method alone cannot realize the effective improvement of image quality.

### 5.2. Experimental Results of Different Finger Vein Image Enhancement Algorithms

As shown in Figure 10, several commonly used finger vein image processing methods are compared with the method in this paper. In order to show the overall effect of the model, a total of six finger vein images with more serious degradation in the SDUMLA-HMT dataset and the ZJ-UVM dataset are selected in this paper. The original six images are shown in Figure 9a.

As shown in Figure 9a, the images are processed by an adaptive histogram equalization algorithm [16]. This method improves the contrast between the venous and nonvenous regions of the finger vein images to some extent. However, some of the vein patterns are not displayed clearly enough or have been lost. Shown in Figure 10b is an image processed using a fusion image descriptor and global histogram equalization method [19]. The method effectively enhances the local details and improves the overall contrast. However, effective segmentation is still not achieved for the finger veins in some severely degraded regions. Shown in Figure 10c is the image after processing with a 2D Gabor filter [23]. Due to the large difference in the gray value thresholds of the different datasets, the image quality is not satisfactory enough when setting fixed parameters for image optimization. Shown in Figure 10d is an adaptive triple Gaussian filter-processed image [24]. The grain of the image becomes clearer to some extent. However, the effect of vein texture segmentation is poor and cannot meet the recognition requirements.

Shown in Figure 9d is the image post-processing with this paper’s method. Compared with the above finger vein image enhancement and restoration algorithms, the degraded vein restoration means proposed in this paper can effectively improve the contrast between the vein region and the non-vein region, and the determination and differentiation of the non-vein region are better. Next, the SDUMLA-HMT and ZJ-UVM datasets are processed using the four methods mentioned above and the method in this paper, respectively, and the processing results are compared. For the convenience of narration, the processed data of the above two datasets are referred to as Pr-SDUMLA-HMT and Pr-ZJ-UVM, respectively.

In order to examine the effectiveness of different methods for finger vein image restoration and enhancement, this experiment was evaluated using mean structural similarity (MSSIM) [41] and the contrast improvement index (CII) [42]. The MSSIM was calculated using the formula:*MSSIM*(*x*, *y*) = *M*^−1^*∑SSIM* (*x_i_*, *y_i_*)(23)
*SSIM*(*x*, *y*) = (2 *ave*(*x*) *ave*(*y*) + *c*) *·* (2 *σ_xy_* + *d*)*/*(*ave*(*x*)^2^ + *ave*(*y*)^2^ + *c*) *·* ((*σ_x_*)^2^ + (*σ_y_*)^2^ + *d*)(24)
where *x* and *y* represent the pre- and post-processed images, respectively. *x_i_* and *y_i_* represent the *i*th local region. *m* represents the number of local regions. *ave* () is the average gray value of the region. *σ_x_* and *σ_y_* are the variance in the region’s gray value. *σ_xy_* is the covariance of the gray value. *c* and *d* are constants. The quality of the original image in this paper is low, so when the MSSIM value is lower, it means a better improvement of the algorithm.

The CII is defined as:*CII* = *C_up_*/*C_raw_*(25)
*C* = *N*^−1^*∑*((*Q_max_* − *Q_min_*)/(*Q_max_* + *Q_min_*))(26)
where *C_up_* and *C_raw_* represent the contrast of the image before and after processing, respectively. *Q_max_* is the maximum value of the gray level of the local region. *Q_min_* is the minimum value of the gray level of the local region. n represents the number of local sub-regions. When the value of CII is higher, the algorithm more effectively improves the image quality.

The four methods mentioned above and the method of this paper were used to process the six images mentioned above (Figure 9a,d, and four images in Figure 10a–d). And the MSSIM and CII values of each image were calculated. The calculated results are shown in Table 1 and Table 2. From the table, it is known that the MSSIM value and CII are optimal for each image in this paper. This shows that the method proposed in this paper can effectively improve the contrast of finger vein images.

Shown in Table 3 are the MSSIM and CII averages for the Pr-SDUMLA-HMT and Pr- ZJ-UVM datasets. The contrast of the images is improved by some amount after processing with the four methods mentioned above and the method in this paper, respectively. Compared to the original image, the MSSIM values of these five methods decreased by 20.8%, 33.5%, 8.7%, 15.8%, and 42%. the CII values were improved by 262.3%, 225.5%, −30.2%, and 307.1%. This shows that this paper is optimal for finger vein image recovery and enhancement.

In order to verify the feasibility and effectiveness of the image recovery and enhancement algorithm proposed in this paper, we use the detection error tradeoff (DET) curve for testing. The DET curve consists of the false acceptance rate (FAR), the correct rejection rate (FRR), and the EER value. The calculation formula is:
*FAR* = *N_A_/N_D_*(27)
*FRR* = *N_R_/N_S_*(28)
where *N_A_* is the number of times the pseudo-finger is recognized as true. *N_D_* is the total number of times the pseudo-finger is recognized. *N_R_* is the number of times the true finger is recognized as false. *N_S_* is the number of times the true finger is recognized.

The EER value is the intersection of the FRR–FAR curve with a straight line *y* = *x*. A smaller EER value indicates that the algorithm has a better overall performance for that user size. In the same situation, the lower the FAR value, the better the security of the algorithm, and the lower the FRR value, the better the passability of the algorithm.

As shown in Figure 11, the DET curves of the results of the Pr-SDUMLA-HMT and Pr-ZJ-UVM datasets’ recognition is shown. We use two classical feature-point matching methods (Sift-Flann [43] and LBP-SVM [44]) to detect the effect of different methods on the image feature-point enhancement. Among them, Figure 11a shows the DET curves of the Sift-Flann image recognition method. Figure 11b is the DET curve of the LBP-SVM image recognition method. From the figure, it can be seen that the EER of the image enhancement algorithm proposed in this paper has better results and the algorithm is more secure and passable under the two means of feature-point extraction and matching recognition.

Shown in Table 4 are the recognition accuracies of the Pr-SDUMLA-HMT and Pr-ZJ-UVM datasets by the Sift-Flann [43], LBP-SVM [44], Resnet50 [45], Goognet [46], and MAML [47] neural network methods. From the table, it can be seen that there are large differences in the enhancement of the image recognition accuracy by the different processing methods, and even for some extraction methods, the processed images are negatively affected. Compared with the other image processing methods mentioned above, the recovery enhancement method based on atmospheric scattering theory proposed in this paper has the most accurate performance of the above methods. The matching accuracy is improved by more than 10% compared to the original image.

### 5.3. Experimental Results of Different Optical Feature-Based Image Enhancement Algorithms

Shown in Figure 12 is a schematic of the undifferentiated images obtained after processing the Pr-SDUMLA-HMT and Pr-ZJ-UVM datasets using the optical feature-based image deblurring algorithm. The comparison shows that the method proposed in this paper gives the best results. The method is able to realize global image recovery and enhancement based on ensuring the contrast of local regions. And it normalizes the gray values of the vein regions and non-vein regions of degraded finger vein images to a certain range. This can effectively improve the accuracy of finger vein recognition.

## 6. Conclusions

Existing finger vein images are subject to severe degradation due to factors such as the performance of the acquisition equipment, the structure of the finger, and the scattering of light. This makes it impossible to quickly and accurately acquire vein features from these images and causes greater interference to finger vein technology recognition. Therefore, this paper proposes a degraded finger vein image restoration and enhancement algorithm based on atmospheric scattering theory. The algorithm is divided into three steps. First, an improved gamma transform method is used to measure the gray values of the image. Then, the image’s gray values are reconstructed based on atmospheric scattering theory. Finally, homomorphic filtering is used to normalize the grayscale of the whole image. Experiments on the SDUMLA-HMT and ZJ-UVM datasets show that the finger vein image restoration and enhancement algorithm proposed in this paper outperforms other commonly used image enhancement algorithms in both MSSIM and CII metrics. In the matching experiments using traditional machine learning and deep learning algorithms, the accuracy of the images processed by this paper’s method is improved by more than 10% compared to the unprocessed images.

In future research, we will continue to explore alternative methods for restoring poor-quality finger vein images. The ultimate goal is to be able to ensure the effective extraction of finger vein features in all special cases and to apply these methods to practical devices.

## Figures and Tables

**Figure 1 sensors-24-02684-f001:**
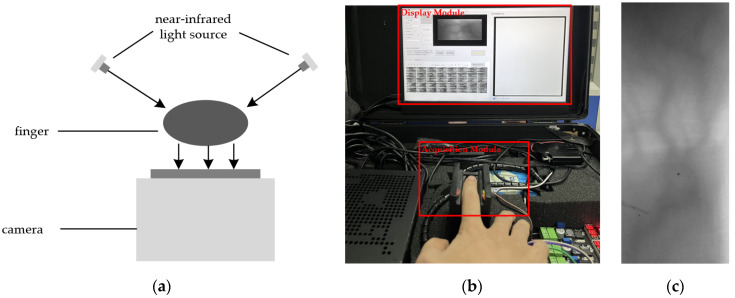
Finger vein image recognition system. (**a**) System schematic. (**b**) Finger vein recognition project device. (**c**) Finger vein image.

**Figure 2 sensors-24-02684-f002:**
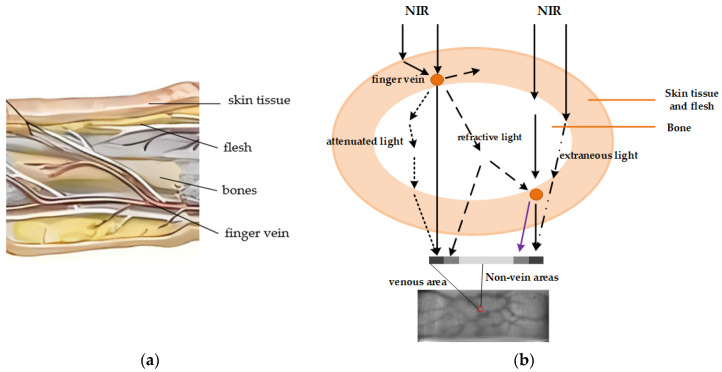
Finger vein scattering model. (**a**) Main internal structures of the fingers. (**b**) Schematic diagram of the scattering effect of near-infrared light inside the finger.

**Figure 3 sensors-24-02684-f003:**
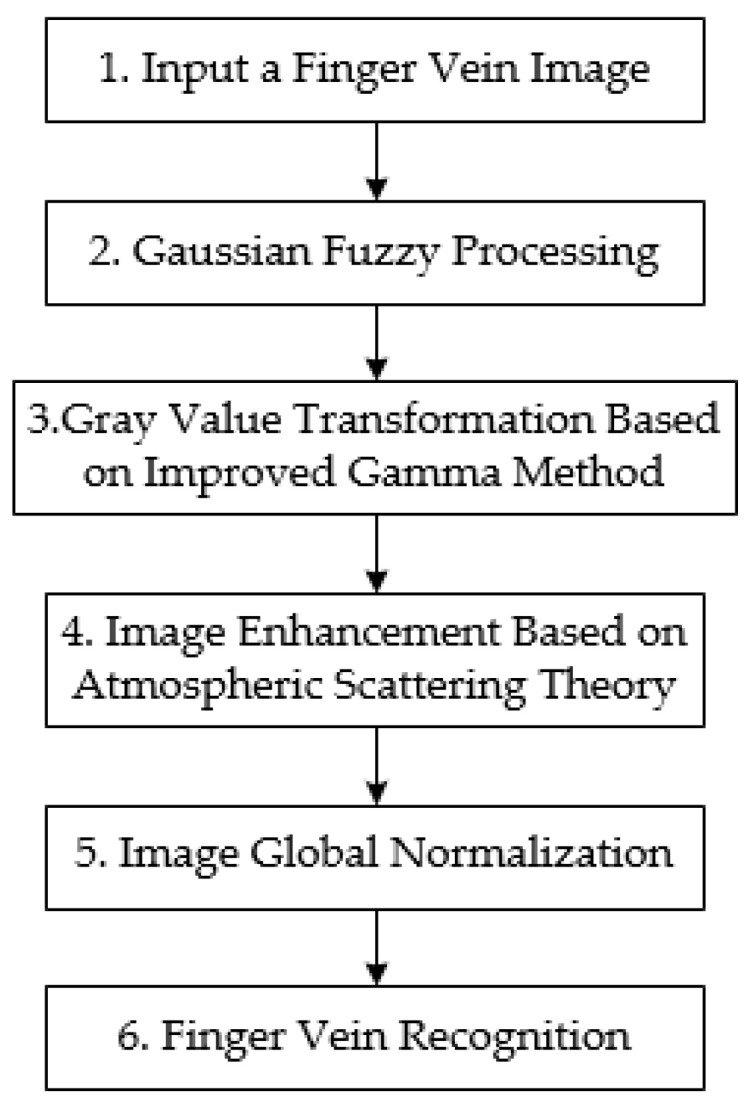
Flowchart of the overall structure of the algorithm in this paper.

**Figure 4 sensors-24-02684-f004:**
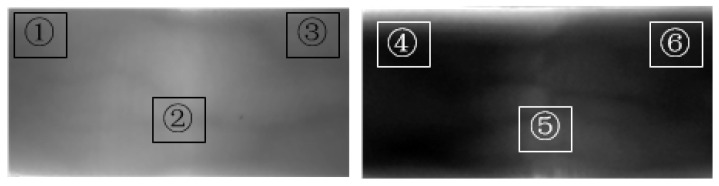
Mean gray values of different regions of the venous image.

**Figure 5 sensors-24-02684-f005:**
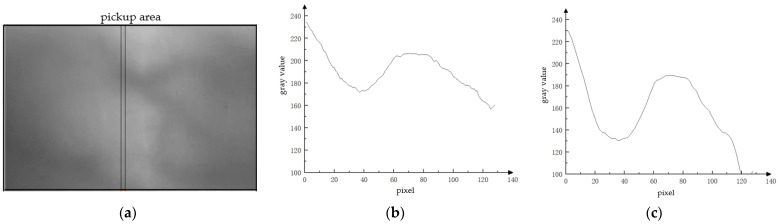
Degraded finger vein image and its gray-value change map. (**a**) Degraded finger vein image I. (**b**) Grayscale value changes in the specified area of image I. (**c**) Grayscale value changes in the specified area of image I after applying atmospheric scattering model restoration. (**d**) Degenerated finger vein image II. (**e**) Change in gray value of the specified area of image II. (**f**) Change in gray value of the specified area of image II after applying atmospheric scattering model restoration.

**Figure 6 sensors-24-02684-f006:**
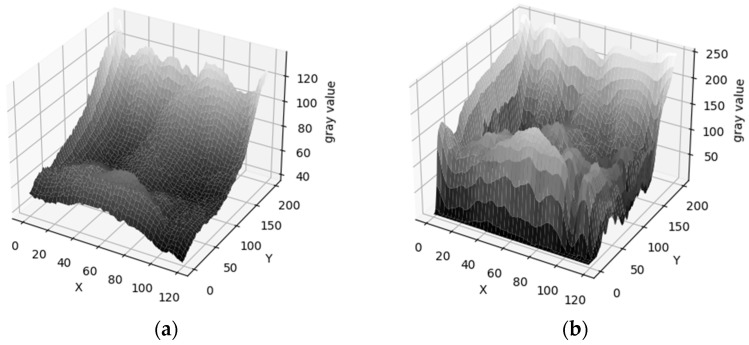
Picture of finger vein image’s gray values. (**a**) Finger vein image III. (**b**) Processed image III. (**c**) Finger vein image IV. (**d**) Processed image IV.

**Figure 7 sensors-24-02684-f007:**
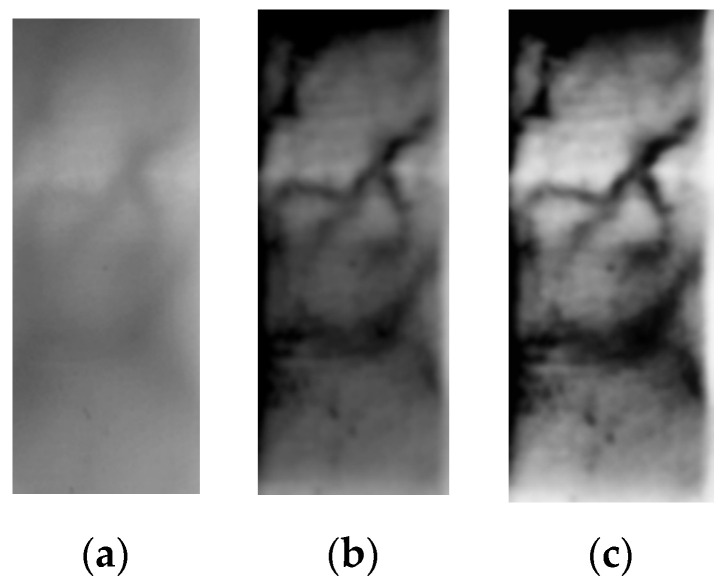
(**a**) Raw finger vein image. (**b**) Picture after processing in Section 4.2. (**c**) Picture after processing in Section 4.3.

**Figure 8 sensors-24-02684-f008:**
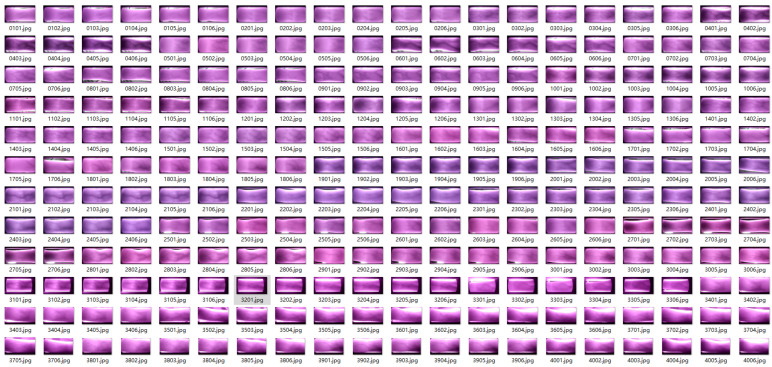
Selected finger vein images from the ZJ-UVM dataset.

**Figure 9 sensors-24-02684-f009:**
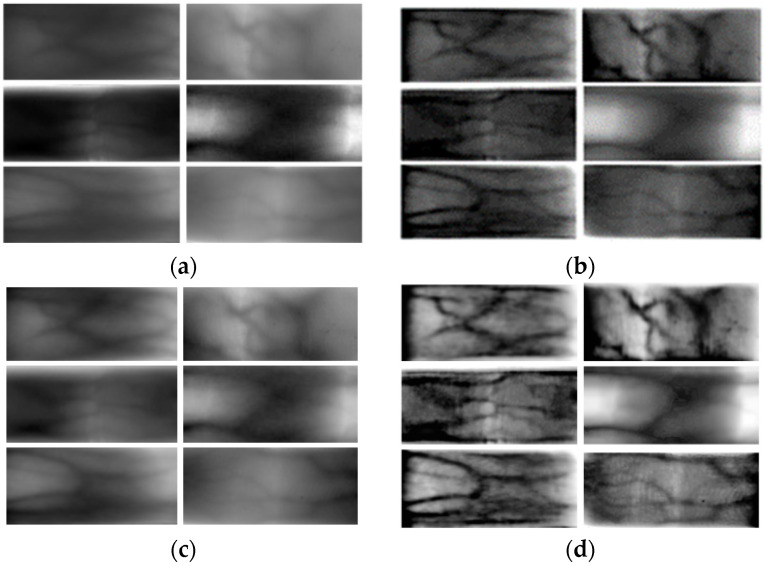
Schematic representation of the results of the ablation experiment. (**a**) Original figure. (**b**) Image after processing by an image enhancement algorithm based on atmospheric scattering theory. (**c**) Image after processing based on homomorphic filtering. (**d**) Image after processing using the method of this paper.

**Figure 10 sensors-24-02684-f010:**
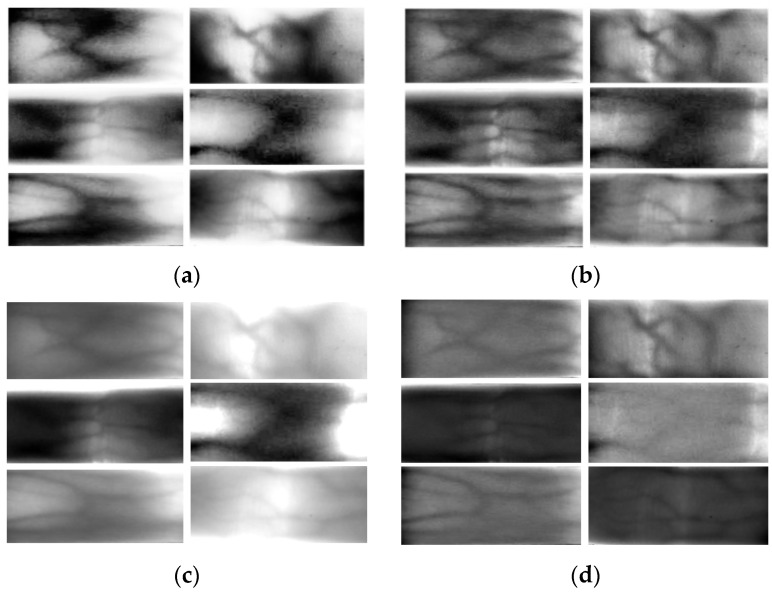
Finger vein images after processing by different image enhancement algorithms. (**a**) Adaptive histogram equalization (CLAHE) [16]. (**b**) Fusion of image descriptors and global histogram equalization [19]. (**c**) 2D Gabor filter [23]. (**d**) Adaptive triple Gaussian filter [24].

**Figure 11 sensors-24-02684-f011:**
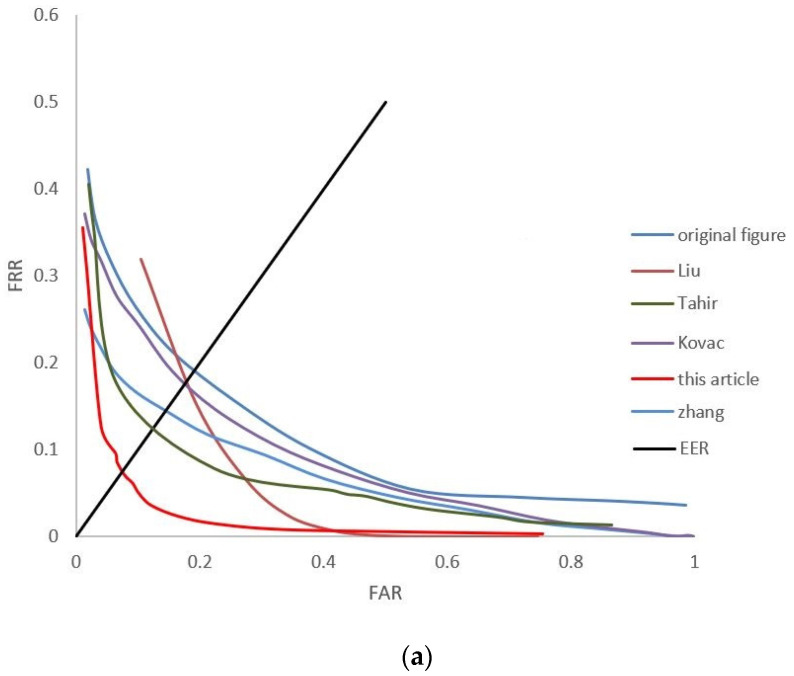
Image recognition DET curve. (**a**) DET curves for Sift-Flann image recognition methods. (**b**) DET curves for LBP-SVM image recognition methods.

**Figure 12 sensors-24-02684-f012:**
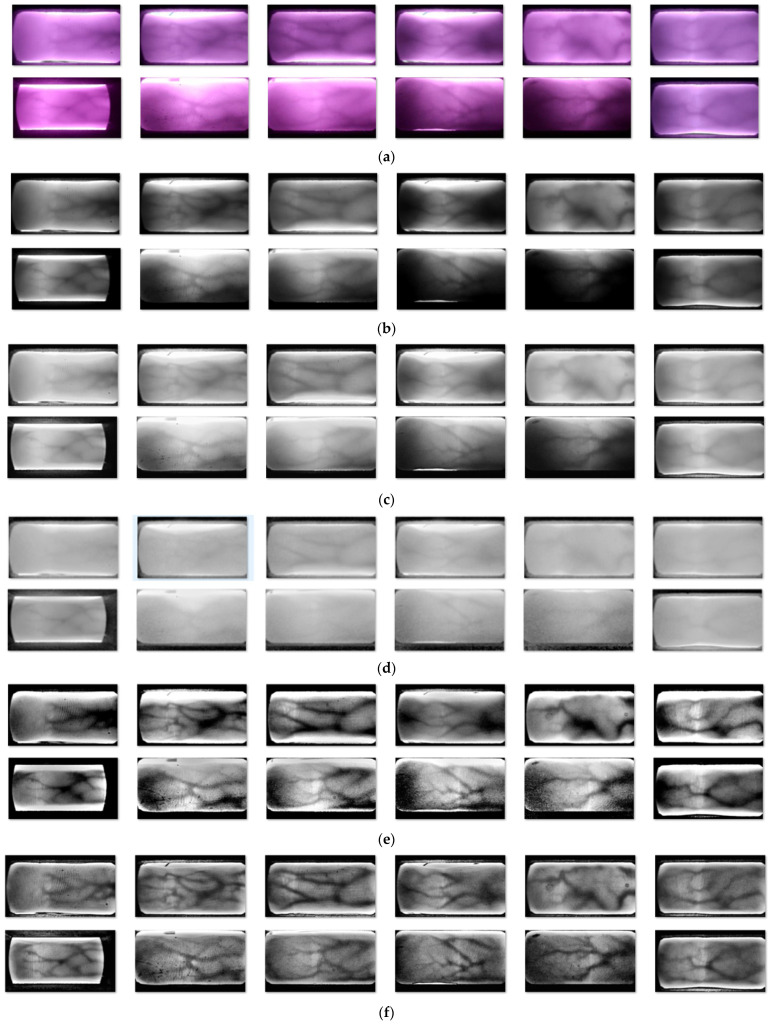
Comparison with other methods. (**a**) Original image. (**b**) Dark-channel a priori algorithm [10]. (**c**) Bi-local adaptive contrast enhancement technique [13]. (**d**) MSRCP [14]. (**e**) AWRM [15]. (**f**) Results of the method in this paper.

**Table 1 sensors-24-02684-t001:** Different image enhancement and restoration algorithms’ MSSIM values.

Method	Image					
	1	2	3	4	5	6
raw image	1	1	1	1	1	1
Liu [16]	0.76	0.88	0.58	0.90	0.74	0.91
Tahir [19]	0.53	0.78	0.48	0.72	0.66	0.83
Kovac [23]	0.84	0.92	0.87	0.91	0.88	0.93
Zhang [24]	0.82	0.87	0.85	0.92	0.78	0.88
this article	0.52	0.60	0.42	0.70	0.47	0.64

**Table 2 sensors-24-02684-t002:** Different image enhancement and restoration algorithmss CII values.

Method	Image					
	1	2	3	4	5	6
raw image	1	1	1	1	1	1
Liu [16]	3.93	4.96	1.40	1.35	4.37	4.73
Tahir [19]	5.05	3.77	1.12	1.25	4.41	3.19
Kovac [23]	0.86	0.47	0.76	0.66	0.78	0.55
Zhang [24]	1.12	2.33	1.45	0.98	2.21	0.89
this article	5.85	4.70	5.72	4.54	8.38	6.59

**Table 3 sensors-24-02684-t003:** Quantitative assessment of the effect of image restoration on the dataset.

Method	MSSIM	CII
raw image	1	1
Liu [16]	0.792	3.623
Tahir [19]	0.665	3.255
Kovac [23]	0.913	0.698
Zhang [24]	0.842	1.322
this article	0.580	4.701

**Table 4 sensors-24-02684-t004:** Accuracy of different image recognition algorithms.

Method	Raw Image	Liu	Tahir	Kovac	Zhang	This Article
Sift_flann	0.735	0.539	0.814	0.547	0.634	0.88
Lbp_svm	0.647	0.726	0.686	0.578	0.633	0.744
Resnet50	0.833	0.804	0.902	0.922	0.786	0.941
Goognet	0.864	0.863	0.961	0.804	0.884	0.966
MAML	0.862	0.861	0.855	0842	0.828	0.926

## Data Availability

The data presented in this study are publicly available in the SDUMLA-HMT, reference number [40]. Additional data are available upon request from the corresponding author. The data is not available to the public due to the privacy of the companies’ data.

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
