# Peer review of "A Degraded Finger Vein Image Recovery and Enhancement Algorithm Based on Atmospheric Scattering Theory"

_sensors, 2024, doi:10.3390/s24092684_

Round 1
Reviewer 1 Report
Comments and Suggestions for Authors
In this paper the authors propose an algorithm to improve finger vein images based on atmospheric scattering theory. The paper provides a rather detailed review of the literature on the subject under study. The authors provide a description of the proposed method and an estimation of its efficiency when processing two datasets. Unfortunately, the quality of the presentation of the results in the paper is very poor. It does not allow us to fully evaluate either the proposed solution itself or its effectiveness.
As remarks to the paper, it is worth mentioning:
1. The first thing that catches my eye is the quality of the English text. I am not an english native speaker, but it was very hard for me to read the text. Here are some examples:
"Given this, this paper..."
"... global picture normalization image algorithm is quoted to drop the gray value ..."
"Secondly, some of the non-finger vein regions in the same image."
"Comparisons with other means."
And even a piece of French text for some reason:
"Zhang Y [10] a proposé un filtre de Gabor adaptatif combiné à un réseau convolutionnel pour améliorer les régions veineuses et supprimer les pseudo-veines to overcome this problem."
It seems that the authors did not even read the article before sending it.
The article also contains a large number of repetitions of the same text in different words. Basically, that finger vein images are of poor quality.
It is also worth noting that a large number of abbreviations have no explanation in the text.
2. The second thing that does not allow to take the work of the authors seriously is very poorly prepared formulas. For the first formula "The total irradiance E received by the sensor is equal to the sum of the attenuation model Edt and the atmospheric light model Eα:". But in formula (1) there are no variables Edt and Eα! The variables used in the formulas are explained in the text of the article for at most 10% of them. What are the differences between formulas (4) and (5), only in the replacement of r(x) by e(x)? In this case e(x) is not transcribed in the text. In formula (7), degrees are not shown. In formula (7) Guess is capitalized, in formula (8) guess is already with a small letter. And so on and so forth. There is not a single correct formula in the article.
3. The text of the paper itself contains too few technical details to evaluate the solution proposed by the authors. For example, Section 2.2 gives an example of processing two images. But it is not clear what the processing is. Is this the processing that the authors propose and describe further in the text? Then it is necessary to move the processing example after the description of the algorithm itself.
What is the meaning of the given images in Figure 1 at "NIR luminance is low"?
Figure 1b appears to be the left half of Figure 1a. What is the meaning of its repetition?
What does the result of processing the figure with "NIR luminance is low" and the graph in Figure 1c show? In my opinion, the processed figure is not suitable for determining vien structure.
The axis captions in figures 1 and 6 are too small.
Figure 2 and its explanation in the text are rather confusing. There is a very similar figure in the article [7], which is much better done. Also, the authors should have added a scheme for obtaining finger vein images similar to the article [7] and given brief explanations in the text.
The article states that the processing algorithm is based on the "Gamma transform" enhancement. It is not clear what the authors mean by this transformation. The authors should provide a reference to the description of the original variant. If it is a grayscale non-linear transformation r = c^gamma, where c is the brightness of a pixel in the original image and r is the brightness of a pixel in the processed image, it is not clear how this algorithm is related to what is described in the article. The authors do not use this transform, at least in the formulas that are given.
It is not clear why pseudocode for homomorphic filtering is given, and is not given for the previous steps in image processing.
4. The description of the performance evaluation of their algorithm also raises questions. Why did the authors choose for comparison the methods described in [9, 28-29], and did not use the work of [7], which is also based on light scattering estimation, or some other techniques described in the literature review?
The DET curve estimation technique is also poorly described. It is not clear what the image recognition technique was and what it involved.
Overall, it is possible that the technical level of the article is sufficient for its publication in a journal, but from the current version of the article it cannot be understood. I believe that the article in its current form cannot be published. And for its publication the authors should completely revise it.
Author Response
Dear Editor,
Thank you for your valuable comments, which have greatly improved the readability of the manuscript. We have revised the manuscript according to your very constructive suggestions and have replied to your comments one by one,for your review.

Reviewer 2 Report
Comments and Suggestions for Authors
The article proposes a finger vein image recovery enhancement algorithm based on atmospheric scattering theory. The algorithm firstly normalizes the over-bright and over-dark regions of the image by gamma transform, then reconstructs the image by using pixel mutation filter according to the atmospheric scattering theory, and finally recovers and enhances the degraded finger vein image by normalizing the gray value of the image. The effectiveness of the method is verified through experiments, but there are some problems at the same time:
1. The methods in related fields introduced in this paper in the "Introduction" are not novel enough, and the authors are requested to add some newer methods. (References and citations can be made: “piecewise color correction and dual prior optimized contrast enhancement”、“Weighted Wavelet Visual Perception Fusion”和“Underwater scene prior inspired”).
2. The article lacks an overall flowchart and needs to be completed by adding the process of running the method in the figure notes.
3. The article lacks a description of equations 1, 2, and 3 and does not annotate the variables that appear in the equations; the authors need to check and revise the entire article.
4. Figures 1, 3, 4, and 5 in the article are not clear, and the authors are requested to provide high-resolution images.
5. The article has too few comparative methods for comparative testing and lacks comparisons with state-of-the-art methods. (Reference can be made to the following methods: “Minimal Color Loss and Locally Adaptive Contrast Enhancement”、 “An Image Restoration Method With Generalized Image Formation Model ”).
6. The article lacks ablation experiments, the authors need to add ablation experiments.
Comments on the Quality of English LanguageThe article proposes a finger vein image recovery enhancement algorithm based on atmospheric scattering theory. The algorithm firstly normalizes the over-bright and over-dark regions of the image by gamma transform, then reconstructs the image by using pixel mutation filter according to the atmospheric scattering theory, and finally recovers and enhances the degraded finger vein image by normalizing the gray value of the image. The effectiveness of the method is verified through experiments, but there are some problems at the same time:
1. The methods in related fields introduced in this paper in the "Introduction" are not novel enough, and the authors are requested to add some newer methods. (References and citations can be made: “piecewise color correction and dual prior optimized contrast enhancement”、“Weighted Wavelet Visual Perception Fusion”和“Underwater scene prior inspired”).
2. The article lacks an overall flowchart and needs to be completed by adding the process of running the method in the figure notes.
3. The article lacks a description of equations 1, 2, and 3 and does not annotate the variables that appear in the equations; the authors need to check and revise the entire article.
4. Figures 1, 3, 4, and 5 in the article are not clear, and the authors are requested to provide high-resolution images.
5. The article has too few comparative methods for comparative testing and lacks comparisons with state-of-the-art methods. (Reference can be made to the following methods: “Minimal Color Loss and Locally Adaptive Contrast Enhancement”、 “An Image Restoration Method With Generalized Image Formation Model ”).
6. The article lacks ablation experiments, the authors need to add ablation experiments.
Author Response

(The authors gave the same response as above.)

Reviewer 3 Report
Comments and Suggestions for Authors
1. The author has written the relevant content of the literature review in the introduction section, rather than elaborating it in a separate section dedicated to the related works section. I recommend modifying the article’s structure.
2. The author did not explain the full names of each abbreviation in the paper (e.g., BOM, DET curve, EER, NIR, IR).
3. There is no mathematical notation or use of italics in the paper, and the superscript or subscript are clearly expressed in the form of equations (Lines 312, 313, and 315 of the manuscript).
4. This manuscript has some problems with the name of the author of the cited literature (e.g.g. Ding S [11], Shand S [19], Gao R [22], He J [21], etc.). (Lines 76, 99, 103, 104 of the manuscript)
5. The literature cited in the paper is mislabeled (e.g.g. Zhang Y [10]), it should be 13. (Line 88 of the paper)
6. The author did not cite the mentioned method (e.g.g. ResNet, GoogNet). (Line 468 of the manuscript).
7. The author did not specify in the paper which database the chart was based on. The results of the experiment were obtained. (Lines 450, 451, 452, 484, 495 of the manuscript).
8. The author mentioned three databases in the experiment, namely FV-USM, SDUMLA-HMT, and ZJ-UVM databases. Among them, although the FV-USM database has been mentioned, the author did not explain the database and clearly stated where its experimental results are located, making it difficult to know whether the database has been adopted by the author. In addition, the author mentioned in the paper that they proposed a new ZJ-UVM database of finger vein images (line 389 of the paper), but the author did not specify how many subjects the database has, and which hardware equipment and external environment it is based on to capture. Therefore, it makes it difficult for me to confirm whether the database really exists and whether he can really be used in this experiment.
10. The authors should recommend making a description with this paper (IEEE and MDPI journals). This manuscript describes some handcraft, deep learning, meta learning methods in recent years. Introduction section: Should cite and explain latest papers on other information securities methods.
[1] Finger Vein Recognition Using DenseNet with a Channel Attention Mechanism and Hybrid Pool, Electronics, vol. 13, no. 3
[2] New hierarchical finger-vein feature extraction method for iVehicles, IEEE Sensors Journal, vol. 22, no. 13, pp. 13612-13621, 2022. (SCI)
[3] FV-MViT: Mobile Vision Transformer for Finger Vein Recognition, Sensors, vol. 24, no. 4
Comments on the Quality of English LanguageThe English writing should be proofread by a professional editor.
Author Response

(The authors gave the same response as above.)
